# 4-Methylumebelliferone Enhances Radiosensitizing Effects of Radioresistant Oral Squamous Cell Carcinoma Cells via Hyaluronan Synthase 3 Suppression

**DOI:** 10.3390/cells11233780

**Published:** 2022-11-25

**Authors:** Kazuki Hasegawa, Ryo Saga, Kentaro Ohuchi, Yoshikazu Kuwahara, Kazuo Tomita, Kazuhiko Okumura, Tomoaki Sato, Manabu Fukumoto, Eichi Tsuruga, Yoichiro Hosokawa

**Affiliations:** 1Department of Radiation Science, Graduate School of Health Sciences, Hirosaki University, Hirosaki 036-8564, Japan; 2Department of Oral and Maxillofacial Surgery, School of Dentistry, Health Sciences University of Hok-Kaido, Tobetsu-cho 061-0293, Japan; 3Department of Radiation Biology and Medicine, Faculty of Medicine, Tohoku Medical and Pharmaceutical University, Sendai 983-8536, Japan; 4Department of Applied Pharmacology, Kagoshima University Graduate School of Medical and Dental Sciences, Kagoshima University, Kagoshima 890-8544, Japan; 5Pathology Informatics Team, RIKEN Center for Advanced Intelligence Project, Tokyo 103-0027, Japan

**Keywords:** radioresistant cells, hyaluronan, oral squamous cell carcinoma, 4-methylumbelliferone, hyaluronan synthase 3, oxidative stress, superoxide dismutase, intracellular hyaluronan, radiosensitization

## Abstract

Radioresistant (RR) cells are poor prognostic factors for tumor recurrence and metastasis after radiotherapy. The hyaluronan (HA) synthesis inhibitor, 4-methylumbelliferone (4-MU), shows anti-tumor and anti-metastatic effects through suppressing HA synthase (HAS) expression in various cancer cells. We previously reported that the administration of 4-MU with X-ray irradiation enhanced radiosensitization. However, an effective sensitizer for radioresistant (RR) cells is yet to be established, and it is unknown whether 4-MU exerts radiosensitizing effects on RR cells. We investigated the radiosensitizing effects of 4-MU in RR cell models. This study revealed that 4-MU enhanced intracellular oxidative stress and suppressed the expression of cluster-of-differentiation (CD)-44 and cancer stem cell (CSC)-like phenotypes. Interestingly, eliminating extracellular HA using HA-degrading enzymes did not cause radiosensitization, whereas HAS3 knockdown using siRNA showed similar effects as 4-MU treatment. These results suggest that 4-MU treatment enhances radiosensitization of RR cells through enhancing oxidative stress and suppressing the CSC-like phenotype. Furthermore, the radiosensitizing mechanisms of 4-MU may involve HAS3 or intracellular HA synthesized by HAS3.

## 1. Introduction

Oral squamous cell carcinoma (OSCC) is a major malignant tumor of the head and neck that represents the 6th most common cancer worldwide and has shown an increasing trend in recent years [1,2]. Besides surgery and chemotherapy, radiotherapy plays an important role in OSCC because it is non-invasive and provides good local control with the recent development of high-precision dose calculation techniques [3,4]. Despite the advances in radiotherapy, the prognosis of patients with OSCC has not improved over the past 30 years [5,6], and the acquisition of radioresistance during fractionated irradiation is considered one of the reasons for poor prognosis [7,8]. In addition to OSCC, some other cancer cells have been reported to exhibit radioresistance during fractionated irradiation, such as non-small-cell lung and prostate cancer cells [9,10]. Radioresistant (RR) cells remain after radiotherapy and cause recurrence and distant metastasis. Thus, they are a major concern associated with poor clinical outcomes [11,12,13]. Therefore, to improve the prognosis of patients with OSCC, it is important to establish strategies to sensitize RR cells.

The hyaluronan (HA) synthesis inhibitor 4-methylumbelliferone (4-MU) reduces the intracellular content of UDP-D-glucuronic acid [14,15,16]. In recent decades, 4-MU has been shown to exert anti-tumor and anti-invasive/-metastatic effects through suppressing HA synthase (HAS) expression in various cancer cells and mouse models [17,18]. Furthermore, our previous studies have indicated that 4-MU treatment with X-ray irradiation promotes anti-inflammatory effects by suppressing interleukin (IL) −6 and −1β and inhibiting intercellular communication involved in anti-oxidant activities [19,20,21]. Elevated expressions of IL-6 and nuclear factor-kappa B (NF-κB), a master regulator of the inflammatory response, have been suggested to enhance resistance to apoptosis and superoxide dismutase (SOD) in cancer cells and induce radioresistance [22,23,24,25]. Based on these findings, 4-MU is a potential radiosensitizer; however, it is unknown whether 4-MU administration radiosensitizes RR cells.

Recently, in vitro RR cell models were established to understand their characteristics. Kuwahara et al. established clinically relevant RR cells via long-term fractionated X-ray irradiation and reported a higher repair capacity of DNA double-strand breaks in these cells compared with that of parental cancer cells [26]. Other studies have reported that RR cells have characteristics similar to those of cancer stem cells (CSCs), are more tumorigenic, and have enhanced anti-oxidant activity [27,28,29]. Therefore, although the mechanisms of radioresistance have been gradually elucidated, the underlying details are unclear, and effective radiosensitizers and clinical strategies are yet to be established.

In this study, we used RR OSCC cell lines established via long-term fractionated X-ray irradiation and investigated, for the first time, the effect of 4-MU as a radiosensitizer.

## 2. Materials and Methods

### 2.1. Reagents

4-MU (Nacalai Tesque, Kyoto, Japan) was diluted in dimethylsulfoxide (DMSO) (Fujifilm Wako Pure Chemical Industries, Ltd., Osaka, Japan) at a working concentration of 500 μM to minimize the cytotoxicity on normal fibroblasts and clearly observe the effects of 4-MU [19]. *Streptomyces* hyaluronidase (*St*-Hyal) (Seikagaku Corporation, Tokyo, Japan) was diluted in dH_2_O and used at a final concentration of 100 TRU/mL. Calcium- and magnesium-free phosphate-buffered saline (PBS (-)) were purchased from Takara Bio Inc. (Otsu, Japan). Monoclonal phycoerythrin (PE)-conjugated anti-human cluster-of-differentiation (CD)-44 antibodies (cat. no. 338808), mouse monoclonal PE-IgG1, κ isotype control (cat. no. 400114), fluorescein isothiocyanate (FITC)-conjugated anti-human CD24 antibodies (cat. no. 311103), and mouse monoclonal FITC-IgG1 κ isotype control (cat. no. 400207) were obtained from BioLegend (San Diego, CA, USA). Small interfering RNA (siRNA) against HAS3 (sc-45295), the corresponding scrambled control siRNA (sc-37007), and anti-HAS3 (sc-365322) monoclonal primary antibodies were purchased from Santa Cruz Biotechnology, Inc. (Santa Cruz, CA, USA). Anti-β-actin (4970) monoclonal antibodies, anti-rabbit horseradish-peroxidase (HRP)-conjugated IgG (7074), and anti-mouse HRP-conjugated IgG (7076) secondary antibodies were purchased from Cell Signaling Technology (Tokyo, Japan). Epidermal growth factor (EGF) and basic fibroblast growth factor (bFGF) were purchased from Fujifilm Wako Pure Chemical, Ltd.

### 2.2. Cell Culture

The human OSCC cell lines HSC2 and HSC3, and their RR counterparts HSC2-R and HSC3-R, were obtained from the Cell Resource Center for Biomedical Research, Institute of Development, Aging and Cancer, Tohoku University. RR cell lines were established via fractionated X-ray irradiation at 2 Gy/day for more than one year [30]. The cell lines were cultured in Roswell Park Memorial Institute (RPMI) 1640 medium (Thermo Fisher Scientific Inc., Waltham, MA, USA), supplemented with 10% heat-inactivated fetal bovine serum (FBS; Japan Bio Serum, Fukuyama, Japan) and 1% penicillin/streptomycin (Fujifilm Wako Pure Chemical Industries, Ltd.) and maintained at 37 °C and in a 5% CO_2_ environment.

### 2.3. Irradiation Condition

The cultured cells were irradiated using an X-ray generator (MBR-1520R-3; Hitachi Medical Co., Tokyo, Japan) as previously reported [20]. The total dose and dose rate of 1.0 Gy/min were measured using an ionizing chamber (MZ-BD-3, Hitachi Medical Co., Tokyo, Japan) placed next to the sample, and the dose rate in the air was determined by converting the air kerma.

### 2.4. SiRNA Transfection

Each cell line was seeded in φ60 mm culture dishes without antibiotics and incubated for 18 h. Samples were washed with PBS (-) and transfected with siRNA using Lipofectamine RNAiMAX (Invitrogen; Thermo Fisher Scientific, Inc.), following the manufacturer’s instructions. The final concentration of both the siRNA and scrambled control siRNA was 50 nM, and the cells were harvested after transfection for 48 h.

### 2.5. HA Density Quantitation

The HA concentration in the culture supernatant was detected using a Hyaluronan Quantikine ELISA kit (R&D Systems, Inc., Minneapolis, MN, USA), as reported previously [31]. The HA concentration was calculated from the standard curve of the absorbance measured at 450 nm.

### 2.6. Clonogenic Survival Assay

The clonogenic potency was evaluated via a colony formation assay. The appropriate number of cells was seeded on φ60 mm culture dishes and incubated for 2 h and subjected to 500 μM 4-MU or 100 TRU/mL *St*-Hyal with 2 Gy X-ray irradiation (IR). Given the cell-killing effects of each treatment, the cells were seeded in different numbers to form the appropriate number of colonies. After treatment for 24 h, 4-MU and *St*-Hyal were washed out, and the cells were further incubated for 7–10 days, fixed with methanol (Fujifilm Wako Pure Chemical Industries) and stained with a Giemsa staining solution (Fujifilm Wako Pure Chemical Industries). Colonies with >50 cells were counted. The surviving fraction for each cell line was calculated from the ratio of the plating efficiency of the irradiated and/or 4-MU- or *St*-Hyal-administrated samples with that of the control samples.

### 2.7. Monolayer Wound Healing Assay

Each cell line was plated and allowed to form a confluent monolayer, which was then scratched with the thin edge of a 200 μL microtip. Cell migration images at 0, 6, 12, and 24 h after treatment were obtained using an Olympus IX71 fluorescence microscope (Tokyo, Japan) and DP2-BSW software (Olympus) at 10× magnification. The wound distance was measured at each time point, and the cell migration rates compared with those at 0 h were calculated. This was performed in three independent experiments.

### 2.8. RNA Extraction and Reverse Transcription-Quantitative Polymerase Chain Reaction (RT-qPCR)

Total RNA was extracted from each sample, and cDNA synthesis was performed with a reaction mixture containing forward and reverse primers, as previously described [20]. RT-qPCR was performed using a real-time PCR system (StepOne Plus; Life Technologies, Waltham, MA, USA) with the following conditions: 95 °C for 30 s, followed by 40 cycles of 95 °C for 5 s and 54 °C for 30 s. Target gene expression levels were calculated relative to glyceraldehyde 3-phosphate dehydrogenase (GAPDH, internal control) mRNA via the comparative ΔΔCq method [32]. The following specific primer sequences were used: HAS1 (forward, 5′-TGTGTATCCTGCATCAGCGGT-3′; reverse, 5′-CTGGAGGTGTACTTGGTAGCATAACC-3′), HAS2 (forward, 5′-CTCCGGGACCACACAGAC-3′; reverse, 5′-TCAGGATACATAGAAACCTCTCACA-3′), HAS3 (forward, 5′-ACCATCGAGATGCTTCGAGT-3′; reverse, 5′-CCATGAGTCGTACTTGTTGAGG-3′), and GAPDH (forward, 5′-GTGAAGGTCGGAGTCAACG-3′; reverse, 5′-TGAGGTCAATGAAGGGGTC-3′).

### 2.9. Flow Cytometric Analysis

To evaluate the expression of HA receptor CD44 and CSC marker CD24, cells were stained with PE-conjugated anti-human CD44 antibodies (3 µL/10^6^ cells) and FITC-conjugated anti-human CD24 antibodies (3 µL/10^6^ cells) and analyzed using FACS Aria Cell Sorter (BD Biosciences, Ltd., Tokyo, Japan) according to previously reported procedures [19]. To perform the appropriate analysis, the following gating strategy was used to define the staining population: the targeted population was gated by forward and side scatter (FSC and SSC, respectively), and the doublets and debris were removed. The gating population was reflected in the dot plot, and 1% each of CD44(+) and CD24(+) were gated in the dot plot as isotype controls. This gate was adapted to each treatment group, and the expressions of CD44 and CD24 were evaluated. Oxidative stress (caused by reactive oxygen species (ROS)), which is intrinsically related to DNA damage induction, was measured via a DCFDA assay (H2DCFDA, Cellular ROS Assay Kit, Abcam, Tokyo, Japan). The mean fluorescence intensity (MFI) of DCFDA per cell was measured at 0, 2, and 24 h after treatments. DCFDA staining and 4-MU administration were simultaneously performed 30 min before irradiation. The increment of the ROS level at 0 h was similar to that observed in our previous report [21]. The gating strategy was similar to that of the CD44 and CD24 analyses; after removal of the doublets and debris gated on the target population based on FSC and SSC dot plots, this gating was reflected in the histogram of each treatment group, and the MFI of DCFDA was evaluated.

### 2.10. SDS-PAGE and Western Blotting

Harvested cells were lysed in 1× radioimmunoprecipitation (RIPA) buffer (Santa Cruz Biotechnology), mixed with 2 × volume gel electrophoresis loading buffer (Tis-Glycine sodium dodecyl sulfate (SDS)) containing 1.5% 2-mercaptoethanol (Fujifilm Wako Pure Chemicals Industries) and boiled at 100 °C for 5 min. The protein concentration was determined using a BCA protein assay kit (Takara Bio) and an iMark microplate reader (Bio-Rad Laboratories, Inc., Hercules, CA, USA). The proteins (~20 μg/lane) were separated using 5–20% EHR-520L e-PAGEL HR (ATTO, Tokyo, Japan) and were electro-transferred onto polyvinylidene difluoride (PVDF) membranes in 25 mM Tris/192 mM glycine, pH 8.3, at 25 V for 2 h. Membranes were blocked using EzBlock Chemi (ATTO) at room temperature for 1 h, and then incubated overnight with primary antibodies, anti-HAS3 (1:1000), and anti-actin antibodies (1:3000) at 4 °C. The membranes were then incubated with HRP-conjugated secondary antibodies in EzBlock Chemi at room temperature for 90 min. The following secondary antibodies were used: HRP-linked anti-rabbit IgG (1:5000) or HRP-linked anti-mouse IgG (1:5000). Antigens were visualized using a chemiluminescence Western blotting substrate (Bio-Rad Laboratories), and blot stripping was performed using a stripping solution (Fujifilm Wako Pure Chemical Industries).

### 2.11. Assay of SOD Activity

SOD activity in each sample was detected using a WST-SOD assay kit (Dojindo Molecular Technologies, Inc., Kumamoto, Japan) according to the manufacturer’s instructions. Cells were harvested in PBS (-) and sonicated with 30 sec pulses (20% output control) on ice. The supernatants were collected after centrifugation of the cell lysate at 10,000× g for 15 min at 4 °C. The supernatants and WST-working solutions were added to each 96-well plate. The enzyme solution was pipetted into each well, and the plates were incubated at 37 °C for 20 min. Subsequently, SOD activity (expressed as unit/10^6^ cells) was measured at 450 nm using an iMark microplate reader.

### 2.12. Statistical Analysis

Data are presented as mean ± SD of the three independent experiments. The one-way analysis of variance and the Tukey–Kramer test were performed to assess the significance of the differences between the control and experimental cultures. Statistical significance was set at *p* < 0.05. Statistical analyses were performed using Microsoft Excel 2016 (Microsoft Corporation, Redmond, WA, USA) with Statcel v4 add-in software (OMS Publishing, Saitama, Japan).

## 3. Results

### 3.1. Evaluation of the Effect of 4-MU on Radioresistance

We first confirmed the efficiency of 4-MU in RR cells. We evaluated the radiosensitization of RR cells treated with 4-MU and IR via a colony formation assay. The surviving fraction of 4-MU-treated cells was significantly lower than that of the control cells, and the combination of 4-MU and IR significantly suppressed the surviving fraction compared with IR alone (Figure 1A). Next, we measured the cell migration ability via a wound healing assay. The cell migration rates were significantly inhibited in HSC2 and HSC2-R cells 12 h after 4-MU treatment and in HSC3 and HSC3-R cells 6 h after treatment compared with those of their controls (Figure 1B,C). We also evaluated cell migration following stimulation with EGF/bFGF. The cell migration rate 12 h after EGF/bFGF treatment was significantly enhanced compared with that of the controls; even under these conditions, 4-MU treatment significantly suppressed cell migration (Figure 1B,C). These results indicated the efficiency of 4-MU in RR cells.

### 3.2. Investigation of Mechanisms of Radiosensitization with 4-MU

Next, we examined the mechanisms by which 4-MU radiosensitizes RR cells. The expressions of CD44 and CD24 (CSC markers) were analyzed via flow cytometry. The ratio of the CD44(+)/CD24(-) fraction, which is used to assess the CSC-like phenotype of OSCC [33,34], significantly decreased after 4-MU treatment compared with that of the controls (Figure 2A,B). In addition, this fraction was suppressed by IR treatment only in the HSC3-R cells. The MFI of CD44 was significantly suppressed through 4-MU treatment compared with that of the controls (Figure 2C). IR treatment showed a similar tendency to suppress the MFI of CD44, and the combination of 4-MU and IR significantly suppressed it compared with that of the controls (Figure 2D). IR treatment did not affect the MFI of CD24, whereas the combination of 4-MU and IR significantly enhanced it compared with IR alone.

As CD44 and CD24 have been reported to contribute to anti-oxidant activity [35,36,37], we evaluated intracellular ROS levels following 4-MU administration. The intracellular ROS levels in HSC2-R and HSC3-R cells were significantly lower than those in their respective parental cells, HSC2 and HSC3 (Figure 3A). The ROS level of the IR group did not change compared to that of the control group, while those of the 4-MU alone and combined groups significantly increased from 0 h, and the effects were sustained for 24 h after treatment (Figure 3B,C). Next, we measured the SOD production levels in the presence of 4-MU. The SOD levels in HSC2-R and HSC3-R cells were significantly higher than those in HSC2 and HSC3 cells (Figure 3D), and IR treatment significantly enhanced SOD activity compared with the control (Figure 3D). The SOD level after 4-MU administration in HSC2-R and HSC3-R cells was significantly suppressed compared with that in their respective control groups, whereas in HSC2 and HSC3 cells, it remained unchanged or slightly increased (Figure 3D). The combination of 4-MU and IR significantly suppressed the SOD levels compared with IR alone. These results suggested that 4-MU administration activates oxidative stress and enhances radiosensitization.

### 3.3. Radiosensitizing Effects after Extracellular HA Elimination or HAS Inhibition

As 4-MU is an HA synthesis inhibitor, we further investigated the radiosensitizing effects of extracellular HA elimination via *St*-Hyal or HAS knockdown by using siRNA. First, as a concentration study for *St*-Hyal, the HA concentration in the cell culture supernatant 24 h after *St*-Hyal treatment was analyzed via ELISA. Because all tested concentrations showed measurement results below the detection limit, we detected the concentration of *St*-Hyal based on the results of other studies [38,39]. Next, we investigated HAS expression and found that HAS1 mRNA expression was not detected, whereas HAS3 expression was significantly higher than HAS2 expression in the RT-qPCR analysis (Appendix A). In addition, HAS3 expression in HSC2-R and HSC3-R cells was much higher than that in HSC2 and HSC3 cells, respectively, and 4-MU administration significantly suppressed HAS3 expression compared with that in the control cells (Appendix A). Therefore, we suppressed HAS3 expression using siRNA, and the knockdown efficiency was confirmed by Western blotting (Figure 4A). As shown in Figure 4A,B, the surviving fraction was suppressed through HAS3 knockdown, whereas it was not affected by *St*-Hyal treatment compared with that in the control group (Figure 4B,C). Similarly, the MFI of CD44 was significantly suppressed through HAS3 knockdown but unaffected by *St*-Hyal treatment (Figure 4D,E). In addition, SOD levels were inhibited through HAS3 knockdown, and co-treatment (HAS3 knockdown and IR) significantly suppressed SOD levels compared with the IR treatment alone (Figure 4F). These results suggested that HAS3 knockdown has similar effects to 4-MU administration and that HAS3 may be an important factor for radiosensitization.

## 4. Discussion

The anti-tumor effects of 4-MU and the role of HAS in various malignancies have been reported [40,41,42]. Our previous studies revealed the enhancement of anti-inflammatory effects and the suppression of anti-oxidant activity by 4-MU [19,20,21]. However, it remains unclear whether 4-MU has radiosensitizing effects on RR cells. Our results demonstrated that 4-MU treatment with IR significantly suppressed cell survival compared with IR alone (Figure 1A), and 4-MU administration significantly suppressed cell migration compared with EGF/bFGF (Figure 1B,C). In addition, 4-MU treatment significantly suppressed the CD44(+)/CD24(-) ratio and MFI of CD44 (Figure 2A–C). In OSCC cell lines, the CD44(+)/CD24(-) fraction has been reported in CSC-like cells [33,34]. CD44 stabilizes xCT, a cystine–glutamate transporter that contributes to glutathione (GSH) synthesis for ROS defense on the cell surface, leading to redox regulation in several cancers [37,43,44]. It was also reported that CD24 contributes to anti-oxidant activity, and its elevated expression enhances oxidative stress in the CD44(+)/CD24(-) phenotype of breast cancer [35]. Moreover, the HA/CD44 interaction was reported to promote inflammatory cytokines [45]; specifically, IL-1β induces human manganese SOD genes such as *SOD2*, which encodes an enzyme that degrades reactive oxygen generated in cells [46]. Our previous study also indicated that 4-MU treatment inhibits the production of IL-1β and IL-6 [19]. The results of this study showed that 4-MU treatment enhanced the intracellular ROS levels and suppressed SOD production (Figure 3). As ROS levels were not affected by IR treatment, the combination of 4-MU and IR showed similar effects as 4-MU treatment alone, whereas SOD production was significantly increased by IR treatment compared with that in the control, and the combination of 4-MU and IR significantly suppressed SOD production compared with IR alone (Figure 3C,D). These results supported the radiosensitizing effects of 4-MU treatment with IR. Furthermore, this study suggested that 4-MU radiosensitizes RR cells and that the mechanisms are based on CSC phenotype suppression and oxidative stress enhancement.

Since 4-MU is an HA synthesis inhibitor, a further upstream mechanistic analysis was performed to investigate its effect on radiosensitization by eliminating extracellular HA using *St*-Hyal or via siRNA-based HAS3 knockdown. Because the HA-binding domain of CD44 is present in the extracellular space, extracellular HA is predominantly used in HA/CD44 interactions [47]. In this study, *St*-Hyal treatment completely removed extracellular HA, but did not affect the radiosensitizing effect and MFI of CD44, whereas HAS3 knockdown by siRNA had effects similar to those of 4-MU treatment (Figure 4). Notably, CD44 expression was not altered via *St*-Hyal treatment but was significantly suppressed via HAS3 knockdown. The HA/CD44 interaction promotes CD44/phosphoinositide 3-kinase (PI3K) complex formation and activates the downstream Akt pathway related to cell survival and anti-apoptosis [48]. Moreover, Akt activation upregulates CD44 expression, which further enhances HA/CD44 interaction (feedback loop) [17]. It was also reported that HAS3 accounts for most of the HA synthesis in OSCC cells [49]. Therefore, it is possible that *St*-Hyal treatment only eliminated extracellular HA but continued to supply HA through HAS, thereby maintaining the anti-oxidant effect of CD44 and the CSC-like phenotype; therefore, it did not lead to a radiosensitizing effect. In addition, 4-MU administration or HAS3 knockdown inhibited HA synthesis and disrupted the feedback loop of the interaction, thus enhancing the radiosensitizing effects.

Given that eliminating extracellular HA did not affect radiosensitization and that HAS releases HA into the intracellular space [50,51,52], it is possible that the intracellular HA synthesized by HAS3 is involved in radiosensitization. The HA synthesized by HAS3 has a low molecular weight (LMW), and the interaction between LMW-HA and CD44 has been reported to promote pro-oncogenic cellular actions [53,54]. Although the HA binding site of CD44 is found in the extracellular space, it was recently reported that CD44 migrates into the nucleus and forms a complex with the signal transducer and activator of transcription 3 (STAT3), which induces cyclin D1 expression and promotes cell proliferation [55]. Therefore, it is possible that intracellular HA and CD44 interact [56]. Furthermore, the receptor for hyaluronan-mediated motility (RHAMM), another major receptor for HA, is known to interact both intracellularly and at the plasma membrane [57,58,59] and was shown to promote tumorigenesis by binding to intracellular HA [60]. Kuo et al. also found that HAS3 and TNF-α form an inter-regulatory loop in oral cancer cells [61] and that TNF-α promotes the binding of NF-κB to the HAS3 promoter region [62]. These findings suggest that interactions with intracellular HA synthesized by HAS3 or the direct action of HAS3 cause radioresistance. However, the role of intracellular HA is not fully understood and requires further investigation.

In conclusion, we showed for the first time that 4-MU treatment is effective in enhancing the radiosensitization of RR cells, suggesting that the sensitization mechanism operates through the suppression of HAS3-mediated oxidative stress enhancement and CSC inhibition.

## Figures and Tables

**Figure 1 cells-11-03780-f001:**
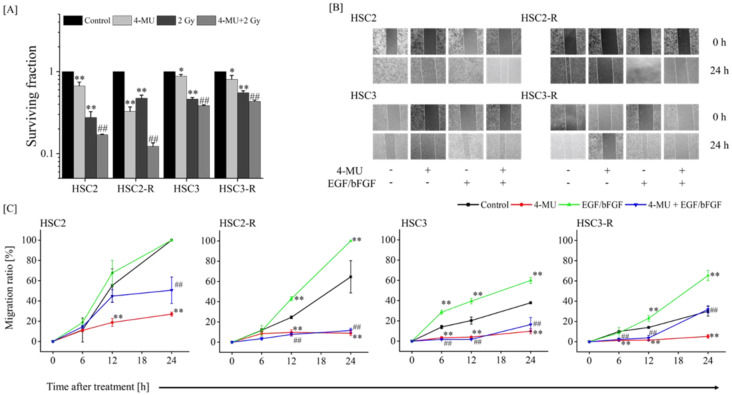
Effects of 4-methylumbelliferone (4-MU) treatment on radioresistant (RR) cells. (**A**) The cell surviving fraction of each cell line under 500 μM 4-MU and 2 Gy X-ray irradiation (IR); * and ** indicate *p* < 0.05 and *p* < 0.01 vs. control, respectively; ^##^ indicates *p* < 0.01 vs. 2 Gy, respectively. (**B**) Cell migration at 0 and 24 h after treatment with 4-MU and epidermal growth factor (EGF)/basic fibroblast growth factor (bFGF) stimulation. (**C**) Cell migration ratio at 0, 6, 12, and 24 h after treatment with 4-MU and EGF/bFGF stimulation; * and ** indicate *p* < 0.05 and *p* < 0.01 vs. control, respectively; ^##^ indicates *p* < 0.01 vs. EGF/bFGF.

**Figure 2 cells-11-03780-f002:**
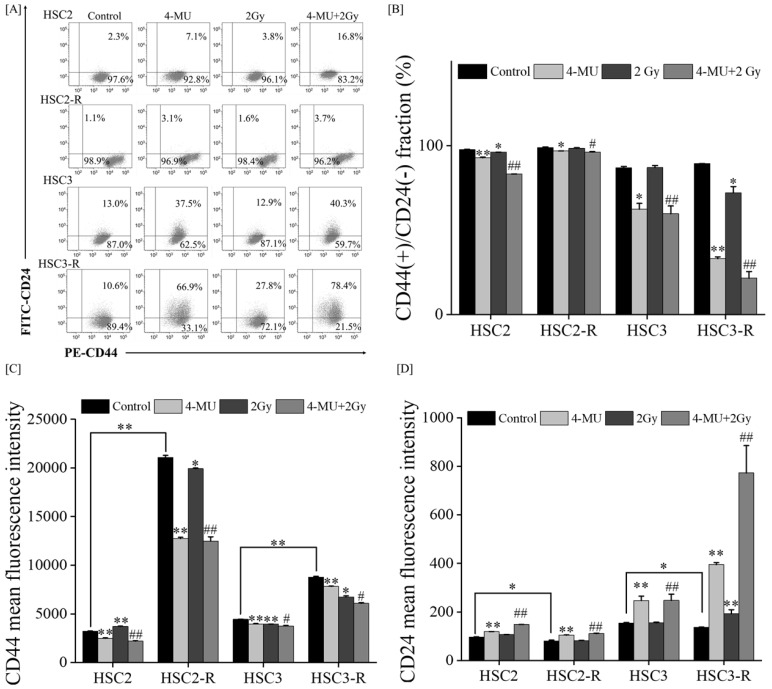
Cluster-of-differentiation (CD)-44 and CD24 expression measured via flow cytometry. (**A**) Representative cytograms of each cell line. (**B**) Cell populations of CD44(+)/CD24(-) irradiated with 2 Gy under 500 μM 4-MU. (**C**,**D**) Mean fluorescence intensity (MFI) of (**C**) CD44 or (**D**) CD24 irradiated with 2 Gy under 500 μM 4-MU; * and ** indicate *p* < 0.05 and *p* < 0.01 vs. control, respectively; ^#^ and ^##^ indicate *p* < 0.05 and *p* < 0.01 vs. 2 Gy, respectively.

**Figure 3 cells-11-03780-f003:**
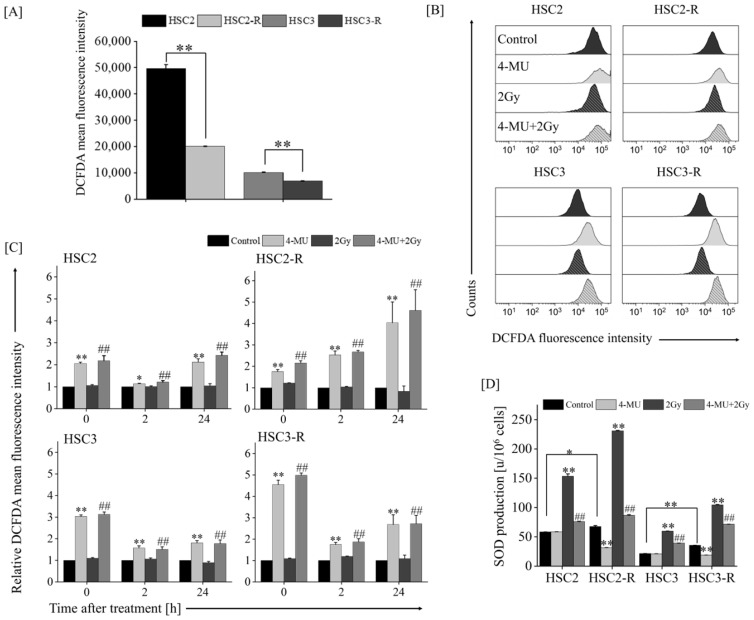
Oxidative stress levels measured via flow cytometry and a superoxide dismutase (SOD) assay. (**A**) Intracellular reactive oxygen species (ROS) levels in each of the control cells. (**B**) Representative histograms of each cell line irradiated with 2 Gy under 500 μM 4-MU. (**C**) Relative intracellular ROS levels in each cell line at 0, 2, and 24 h after treatment. The MFI of IR alone, 500 μM 4-MU alone, and the combination (4-MU + IR) were standardized based on the MFI of the control group at each time. (**D**) SOD production levels of each cell line after treatment for 24 h; * and ** indicate *p* < 0.05 and *p* < 0.01 vs. control, respectively; ^##^ indicates *p* < 0.01 vs. 2 Gy.

**Figure 4 cells-11-03780-f004:**
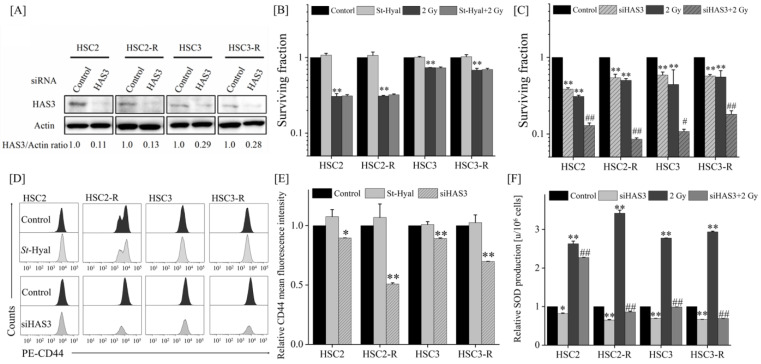
Effects of *St*-Hyal treatment or HAS3 knockdown on RR cells. (**A**) Protein expression analysis of siRNA-HAS3 transfection via Western blotting. Representative images of immunoblots are shown. Actin was used as a loading control, and the relative values of the HAS3/actin ratio are presented. For the HAS3 proteins, both bands were quantified together. (**B**,**C**) Logarithmic surviving fraction of each cell line treated with (**B**) 100 TRU/mL *St*-Hyal and IR or (**C**) HAS3 knockdown and IR. (**D**) Representative histograms of each cell line treated with 100 TRU/mL *St*-Hyal or with HAS3 knockdown. (**E**) Relative MFI of CD44 of each cell line treated with 100 TRU/mL *St*-Hyal or with HAS3 knockdown. (**F**) Relative SOD production levels of each cell line treated with HAS3 knockdown and IR. The SOD production levels of IR alone, HAS3 knockdown, and combined treatment groups were standardized based on the SOD level of the control group; * and ** indicate *p* < 0.05 and *p* < 0.01 vs. control, respectively; ^#^ and ^##^ indicate *p* < 0.05 and 0.01 vs. 2 Gy, respectively.

## Data Availability

The data presented in this study are available in this article.

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
