# Peer review of "4-Methylumebelliferone Enhances Radiosensitizing Effects of Radioresistant Oral Squamous Cell Carcinoma Cells via Hyaluronan Synthase 3 Suppression"

_cells, 2022, doi:10.3390/cells11233780_

Round 1

Reviewer 1 Report

In the current study, Kazuky Hasegawa and colleagues studied in-vitro the effect of the hyaluronan synthetase inhibitor 4-methylumebelliferone on the radiation sensitivity of 2 radioresistant human squamous cell carcinoma cell lines. They demonstrated that HA synthetase by inhibition can revert the radio resistance or RR OSCC, potentially by decreasing the expression of CD44 and increasing oxidative stress.  The statistical analysis seem adequate. I have several comments and suggestions to the authors in order to clarify some points of the study.

Major Concerns:

·        As the authors focused on OSCC cell lines, the title should mention the focus of the study on oral squamous cell carcinoma and should not sound like a general statement on radio resistance of cancer cells.

·   The focus and the rational to focus on OSCC should be explained in the introduction and explained the challenge of radio resistance in OSCC radiotherapy treatment.

·        The description of the results is insufficient for most of the figure, lacking a complete description of the data or a clear explanation of the time of sampling and measurement

·         Figure1

 How was the concentration of 500uM for 4-MU treatment chosen? How this concentration compared with other study

·        (B) Is this really due to migration inhibition or proliferation inhibition. Analysis of cell cycle should be carried out to rule out that the scratch repopulation inhibition is not due to cell cycle inhibition

Figure2:

·        How long after treatment was the flow cytometry evaluation of CD44 and CD24 done?

·        No explanation on the gating strategy and according to the material and methods no cell death marker was used to exclude potential dead cells from the analysis. As 4-MU alone was shown to decrease cell survival in figure one, dead cell present in the analysis could bias the MFI value of surface markers such as CD44/CD24

·        How were the CD44+ and CD24+ population define? Did the authors used single antibody staining and FMO? As the authors measure MFI ensuring that the flow cytometry plot gates to define positive population are correctly set is a must.

·        The description of the results in Fig2(B) and (C) need to be ameliorated, no description of the effect of radiation with or without 4-MU is described. The general statement made on CD44 or CD24 MFI is not descriptive enough.

·        Combination of 4-MU and radiation does not make any difference in comparison to 4-MU alone, thus it seems that 4-MU alone is sufficient to explain the differences seen in ROS level. This should be made clear and discuss in parallel of the results seen for SOD levels. An interesting result is that 4-MU is able to suppress the stimulation of SOD expression by radiation. This might explain some of the survival differences seen by the combination of 4-MU and irradiation.

Figure3:

·        Gating strategy for flow analysis? Exclusion of dead cells?

·        The description of the results is insufficient and doesn’t seem to be sustained by the data concerning the IR group. The author mention that the ROS level at 2hr of the IR group decreased to the level of control group, but no differences in ROS level in IR group seems to happen in any of the time point depicted in frig3(C).

·        How can the ROS level of each cell lines in 4-MU or 4-MU+IR group be already increase at 0hr after treatment? Do the authors pretreated the cells with 4-MU before irradiation, and if yes for how long.

Figure4:

·        Same remark as before for the flow cytometry data (gating strategy, dead cells….)

·        The authors mention that the efficient removal of HA from the culture medium was verified by ELISA, but no results are shown. HA ELISA results should be depicted in supplementary figures.

Discussion:

See comments on figure for additional discussion needed.

Minor comments:

·        Fig1 C: Y-axis legend should be “Cell migration ratio” and not “The cell migration ration”

·        Fig1 D: Y-axis legend should be “CD24” and not “CD44”

·        Fig3 A-B-C: The axis should not mention “ROS” but instead DCFDA as the fluorescence measured is based on the signal emitted by DCFDA in presence of ROS

·        Grammar and spelling throughout the manuscript should be checked

Reviewer 2 Report

In the manuscript entitled "4-methylumebelliferone enhances radiosensitization of radioresistant cells via hyaluronan synthase 3", Kazuki Hasegawa et al. investigated the anticancer and radiosensitizing potential of 4-methylumebelliferone using head and neck cancer cell lines. The authors performed the experiments both on the parental line and on the radioresistant counterpart, the failure model of radiotherapy. The problem of the failure of radiotherapy and the possible consequences in terms of local and distant relapse is clinically very relevant. In clinical practice, re-irradiation for recurrent head and neck cancer is not always a viable practice due to toxicity. The use of radiosensitizers could ensure re-irradiation, using lower radiation doses. The authors demonstrate how 4-methylumebelliferone possesses this characteristic and how it acts by targeting the cancer stem population and increasing intracellular oxidative stress. In my opinion, however, the authors should characterize the ultimate downstream of the radiosensitizing action and therefore the DNA damage and repair. For this I strongly recommend the evaluation of H2AX, DNA-PKCs and phosphorylated ATM.

Round 2

Reviewer 1 Report

Thank you for adressing the comments of the previous review. The paper reads good and is now more comprehnesible and detailled. In my opinion the paper is ready for publication once the following text is corrected:  

1- Please correct figure1C by removing "The" on the Y-axis and just keep "Migration ratio"

2- Please correct figure3D by removing "FITC-" on the X-axis and just keep "DCFDA fluorescence intensity"
